# Disentangled cyclic reconstruction for domain adaptation

## Abstract

The domain adaptation problem involves learning a unique classification or regression model capable of performing on both a source and a target domain. Although the labels for the source data are available during training, the labels in the target domain are unknown. An effective way to tackle this problem lies in extracting insightful features invariant to the source and target domains. In this work, we propose splitting the information for each domain into a task-related representation and its complimentary context representation. We propose an original method to disentangle these two representations in the single-domain supervised case. We then adapt this method to the unsupervised domain adaptation problem. In particular, our method allows disentanglement in the target domain, despite the absence of training labels. This enables the isolation of task-specific information from both domains and a projection into a common representation. The task-specific representation allows efficient transfer of knowledge acquired from the source domain to the target domain. We validate the proposed method on several classical domain adaptation benchmarks and illustrate the benefits of disentanglement for domain adaptation.

## 1 Introduction

The wide adoption of Deep Neural Networks in practical supervised learning applications is hindered by their sensitivity to the training data distribution. This problem, known as *domain shift*, can drastically weaken, in real-life operating conditions, the performance of a model that seemed perfectly efficient in simulation. Learning a model with the goal of making it robust to a specific domain shift is called *domain adaptation* (DA). Often, the data available to achieve DA consist of a labeled training set from a source domain and an unlabeled sample set from a target domain. This yields the problem of *unsupervised domain adaptation* (UDA).

In this work, we take an information disentanglement perspective on UDA. We argue that a key to efficient UDA lies in separating the necessary information to complete the network's task (classification or regression), from a task-orthogonal information which we call context or *style*. Disentanglement in the target domain seems however a difficult endeavor since the available data is unlabeled. Our contribution is two-fold. We propose a formal definition of the disentanglement problem for UDA which, to the best of our knowledge, is new. Then we design a new learning method, called DiCyR (Disentangled Cyclic Reconstruction), which relies on cyclic reconstruction of inputs in order to achieve efficient disentanglement, including in the target domain. We derive DiCyR both in the supervised learning and in the UDA cases.

This paper is organized as follows. Section 2 presents the required background on supervised learning and UDA, and proposes a definition of disentanglement for UDA. Section 3 reviews recent work in the literature that allow for a critical look at our contribution and put it in perspective. Section 4 introduces DiCyR, first for the single-domain supervised learning case and then for the UDA problem. Finally, Section 5 empirically evaluates DiCyR against state-of-the-art methods and discusses its strengths, weaknesses and variants. Section 6 summarizes and concludes this paper.

## 2 PROBLEM DEFINITION

In this section, we introduce the notations and background upon which we build the contributions of Section 4. Let $\mathcal{X}$ be an input space of descriptors and $\mathcal{Y}$ an output space of labels. A supervised learning problem is defined by a distribution $p_s(x, y)$ over elements of $\mathcal{X} \times \mathcal{Y}$. In what follows, $p_s$ will be called the *source* distribution. One wishes to estimate a mapping $\hat{f}$ that minimizes a loss function of the form $\mathbb{E}_{(x,y) \sim p_s} \left[ l(\hat{f}(x), y) \right]$. The optimal estimator is denoted $f$ and one often writes the distribution $\mathbb{P}(y|x)$ as $y \sim f(x) + \eta$, where $\eta$ captures the deviations between $y$ and $f(x)$. Hence, one tries to learn $f$. In practice, the loss can only be approximated using a finite set of samples $\{(x_i, y_i)\}_{i=1}^{n}$ all independently drawn from $p_s$ and $\hat{f}$ is a parametric function (such as a deep neural network) of the form $y = \hat{f}(x; \theta)$.

Domain adaptation (DA) consists in considering a *target* distribution $p_t$ over $\mathcal{X} \times \mathcal{Y}$ that differs from $p_s$, and the transfer of knowledge from learning in the source domain ($p_s$) to the target domain ($p_t$). Specifically, *unsupervised* DA exploits the knowledge of a labelled training set $\{(x_i^s, y_i^s)\}_{i=1}^{n}$ sampled according to $p_s$, and an unlabelled data set $\{(x_i^t)\}_{i=1}^{m}$ sampled according to $p_t$. For instance, the source domain data could be a set of labelled photographs of faces, and the target domain data, a set of unlabelled face photographs, taken with a different camera under different exposure conditions. The problem consists in minimizing the target loss $\mathbb{E}_{(x,y) \sim p_t} \left[ l(\hat{f}(x), y) \right]$.

We suppose that a necessary condition to benefit from the knowledge available in the source domain and transfer it to the target domain is the existence of a common information manifold between domains, where an input's projection is sufficient to predict the labels. We call this useful information *task-specific* or *task-related*. The complimentary information should be called *task-orthogonal*; it is composed of information that is present in the input but is not relevant to the task at hand. For the sake of naming simplicity, we will call this information *style*. However we insist that this should not be confused with the classical notion of style.

Let $\Pi_\tau : \mathcal{X} \to \mathcal{T}$ and $\Pi_\sigma : \mathcal{X} \to \mathcal{S}$ denote two projection operators, where $\mathcal{T}$ and $\mathcal{S}$ denote respectively the latent task-related information space and the latent style-related information space. Let $\Pi$ be the joint projection $\Pi(x) = (\Pi_\tau(x), \Pi_\sigma(x))$. Conversely, we shall note $\bar{\Pi} : \mathcal{T} \times \mathcal{S} \to \mathcal{X}$ a reconstruction operator. And finally, $c : \mathcal{T} \to \mathcal{Y}$ will denote the labeling operator which only uses information from $\mathcal{T}$. We consider that the information of the elements of $\mathcal{X}$ is correctly disentangled by $\Pi = (\Pi_\tau, \Pi_\sigma)$ if one can find $\bar{\Pi}$ and $c$ such that:

C1: $c \circ \Pi_\tau$ minimizes the loss (and thus fits $f$ on the appropriate domain),

C2: $\bar{\Pi} \circ \Pi$ fits the identity operator $id_\mathcal{X}$,

C3: With $X, T, S$ the random variables in $\mathcal{X}, \mathcal{T}, \mathcal{S}$, the mutual information $I(T, S|X) = 0$,

C4: There is no function $g : \mathcal{T} \to \mathcal{X}$ such that $g \circ \Pi_\tau = id_\mathcal{X}$,

Condition C1 imposes that the projection into $\mathcal{T}$ retains enough information to correctly label samples. Condition C2 imposes that all the information necessary for the reconstruction is preserved by the separation performed by $\Pi$. Condition C3 states that no information is present in both $\mathcal{T}$ and $\mathcal{S}$. Condition C4 impose that the information contained in $\mathcal{T}$ alone is insufficient to reconstruct an input, and thus the information of $\mathcal{S}$ is necessary. Note that the symmetrical condition is unnecessary, since the combination of C1 and C3 already guarantees that $\mathcal{S}$ cannot contain the task-related information. Overall, solving this disentanglement problem for DA implies finding a quadruplet $\langle \Pi_\tau, \Pi_\sigma, \bar{\Pi}, c \rangle$ that meets the conditions above. In particular, note that conditions C3 and C4 open a perspective to a formulation of disentanglement in the general case.

## 3 RELATED WORK

Disentanglement between the domain-invariant, task-related information and the domain-specific, task-orthogonal, style information is a desirable property to have for DA. In the next paragraphs, we cover important work in representation disentanglement, domain adaptation and their interplay.

Before deep learning became popular, Tenenbaum & Freeman (2000) presented a method using bi-linear models able to separate style from content. More recently, methods based on generative

models have demonstrated the ability to disentangle factors of variations from elements of a single domain (Rifai et al., 2012; Mathieu et al., 2016; Chen et al., 2016; Higgins et al., 2017; Sanchez et al., 2019). In a cross-domain setting, Gonzalez-Garcia et al. (2018) use pairs of images with the same labels from different domains to separate representations into a shared information common to both domains and a domain-exclusive information. We note that these approaches do not explicitly aim at respecting all conditions listed in Section 2. Additionally, most require labeled datasets (and in some cases even paired datasets) and thus do not address the *unsupervised* DA problem.

One approach to UDA consists in aligning the source and target distributions statistics, a topic closely related to *batch normalization* (Ioffe & Szegedy, 2015). Sun et al. (2017) minimize the distance between the covariance matrices of the features extracted from the source and target domains. Assuming the domain-specific information is contained inside the batch normalization layers, Li et al. (2017) align the batch statistics by adopting a specific normalization for each domain. Cariucci et al. (2017) aim to align source and target feature distributions to a reference one and introduce domain alignment layers to automatically learn the degree of feature alignment needed at different levels of the network. Similarly, Roy et al. (2019) replace batch normalization layers with domain alignment layers implementing a so-called feature whitening. A major asset of these methods is the possibility to be used jointly with other DA methods (including the one we propose in Section 4). These methods jointly learn a common representation for elements from both domains. Conversely, Liang et al. (2020) freeze the representations learned in the source domain before training a target-specific encoder to align the representations of the target elements by maximizing the mutual information between intermediate feature representations and outputs of the classifier.

Ensemble methods have also been applied to UDA (Laine & Aila, 2017; Tarvainen & Valpola, 2017). French et al. (2018) combine stochastic data augmentation with self-ensembling to minimize the prediction differences between a student and a teacher network in the target domain.

Another approach involves learning domain-invariant features, that do not allow to discriminate whether a sample belongs to the source or target domain, while still permitting accurate labeling in the source domain. This approach relies on the assumption that such features allow efficient labeling in the target domain. Ghifary et al. (2016) build a two-headed network sharing common layers; one head performs classification in the source domain, while the second is a decoder that performs reconstruction for target domain elements. Ganin et al. (2016) propose the DANN method and introduce Gradient Reversal Layers to connect a domain discriminator and a feature extractor. These layers invert the gradient sign during back-propagation so that the feature extractor is trained to fool the domain discriminator. Shen et al. (2018) modify DANN and replace the domain discriminator by a network that approximates the Wasserstein distance between domains. Tzeng et al. (2017) optimize, in an adversarial setting, a generator and a discriminator with an inverted label loss.

Other methods focus on explicitly disentangling an information shared between domains (analogous to the domain-invariant features above) from a domain-specific information. Inspired by Chen et al. (2016), Liu et al. (2018b) isolate a latent factor, representing the domain information, from the rest of an encoding, by maximizing the mutual information between generated images and this latent factor. Some domain information may still be present in the remaining part of the encoding and thus may not comply with conditions C3 and C4. Liu et al. (2018a) combine an encoder, an image generator, a domain discriminator, and a fake images discriminator to produce cross-domain images. The encoder is trained jointly with the domain discriminator to produce domain-invariant representations. Li et al. (2020) disentangle a latent representation into a global code and a local code. The global code captures category information via an encoder with a prior, and the local code is transferable across domains, which captures the style-related information via an implicit decoder. Bousmalis et al. (2016) also produce domain-invariant features by training a shared encoder to fool a domain discriminator. They train two domain-private encoders with a difference loss that encourages orthogonality between the shared and the private representations (similarly to condition C3). Cao et al. (2018); Cai et al. (2019); Peng et al. (2019) combine a domain discriminator with an adversarial classifier to separate the information shared between domains from the domain-specific information. All these methods build a shared representation that prevents discriminating between source and target domains, while retaining enough information to correctly label samples from the source domain. However, because they rely on an adversarial classifier that requires labeled data, they do not guarantee that the complimentary, domain-specific information for samples *in the target domain* does not contain information that overlaps with the shared representation. In other words,

they only enforce C3 in the source domain. They rely on the assumption that the disentanglement will still hold when applied on target domain elements, which might not be true.

Another identified weakness in methods that achieve a domain-invariant feature space is that their representations might not allow for accurate labeling in the target domain. Indeed, feature alignment does not necessarily imply a correct mapping between domains. To illustrate this point, consider a binary classification problem (classes $c_1$ and $c_2$) and two domains ($d_1$ and $d_2$). Let $(c_1, d_1)$ denote samples of class $c_1$ in $d_1$. It is possible to construct an encoding that projects $(c_1, d_1)$ and $(c_2, d_2)$ to the same feature values. The same holds for $(c_1, d_2)$ and $(c_2, d_1)$ for different feature values. This encoding allows discriminating between classes in $d_1$. It also fools a domain discriminator since it does not allow predicting the original domain of a projected element. However, applying the classification function learned on $d_1$ to the projected $d_2$ elements leads to catastrophic predictions. Transforming a sample from one domain to the other, while retaining its label information can be accomplished by image-to-image translation methods. Hoffman et al. (2018) extend CycleGAN's cycle consistency (Zhu et al., 2017) with a semantic consistency to translate from source to target domains. The translated images from the source domain to the target domain are then used to train a classifier on the target domain using the source labels. Similarly, Russo et al. (2018) train two conditional GANs (Mirza & Osindero, 2014) to learn bi-directional image mappings constrained by a class consistency loss and use a source domain classifier to produce pseudo-labels on source-like transformed target samples. By relaxing CycleGAN's cycle consistency constraint and integrating the discriminator in the training phase, Hosseini-Asl et al. (2019) address the DA problem in the specific setting where the number of target samples is limited. Takahashi et al. (2020) use a CycleGAN to generate cross-domain pseudo-pairs and train two domain-specific encoders to align features extracted from each pseudo-pair in the feature space. A major asset of the method is to address the class-unbalanced UDA problem by oversampling with the learned data augmentation. Yang et al. (2019) use separate encoders to produce domain-invariant and domain-specific features in both domains. They jointly train these encoders with two generators to produce cross-domain elements able to fool domain-specific discriminators. Using a cyclic loss on features, they force the information contained in the representation to be preserved during the generation of cross-domain elements. However, the cyclic loss on features does not prevent the information sharing between features expressed in C3. More importantly it does not prevent the domain-specific features to be constant. A major drawback of these methods lies in the instability during training that might be caused by min-max optimization problem induced by the adversarial training of generators and discriminators.

In the next section, we introduce a method that does not rely on a domain discriminator and an adversarial label predictor, but directly minimizes the information sharing between representations. This allows to guarantee that there is no information redundancy between the task-related and the task-orthogonal style information in both the source and the target domains. Along the way, it provides an efficient mechanism to disentangle the task-related information from the style information in the single domain case. Our method combines information disentanglement, intra-domain and cross-domain cyclic consistency to enforce a more principled mapping between each domain.

## 4 Disentanglement with Gradient Reversal Layers and cyclic reconstruction

First, we propose an original method to disentangle the task-related information from the style information for a single domain in a supervised learning setting. In a second step, we propose an adaptation of this method to learn these disentangled representations in both domains for UDA. This disentanglement allows, in turn, to efficiently predict labels in the target domain.

### 4.1 Task-style disentanglement in the supervised case

Our approach consists in estimating jointly $\Pi$, $\bar{\Pi}$ and $c$ as a deep feed-forward neural network. We shall note $\theta_\Pi$, $\theta_{\bar{\Pi}}$, and $\theta_c$ the parameters of the respective sub-parts of the network. $\Pi \circ \bar{\Pi}$ takes the form of an auto-encoder, while $\Pi \circ c$ is a task-related (classification or regression) network. Figure 1a summarizes the global architecture which we detail in the following paragraphs.

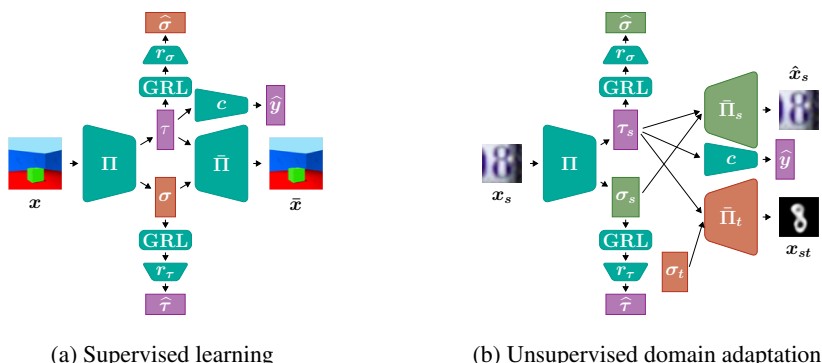

(a) Supervised learning          (b) Unsupervised domain adaptation

Figure 1: Network architectures

Conditions C1 and C2 are expressed through the definition of a task-specific loss $\mathcal{L}_{task}$ (*e.g.* cross-entropy for classification, L2 loss for regression) and a reconstruction loss $\mathcal{L}_{reco}$. Therefore, the update of $\theta_\Pi$ should follow $-\nabla_{\theta_\Pi}(\mathcal{L}_{task} + \mathcal{L}_{reco})$, the update of $\theta_{\bar{\Pi}}$ relies on $-\nabla_{\theta_{\bar{\Pi}}}\mathcal{L}_{reco}$, and that of $\theta_c$ uses $-\nabla_{\theta_c}\mathcal{L}_{task}$.

In order to achieve condition C3, we exploit Gradient Reversal Layers (Ganin et al., 2016, GRL). We train two side networks $r_\tau : \mathcal{S} \to \mathcal{T}$ and $r_\sigma : \mathcal{T} \to \mathcal{S}$ whose purpose is to attempt to predict $T$ given $S$, and $S$ given $T$ respectively. For a given $x$, let us write $(\tau, \sigma) = \Pi(x)$, $\widehat{\tau} = r_\tau(\sigma)$, and $\widehat{\sigma} = r_\sigma(\tau)$. We train $r_\tau$ and $r_\sigma$ to minimize the losses $\mathcal{L}_{r_\tau} = \|\tau - \widehat{\tau}\|_2$ and $\mathcal{L}_{r_\sigma} = \|\sigma - \widehat{\sigma}\|_2$. Let $\mathcal{L}_{info} = \mathcal{L}_{r_\tau} + \mathcal{L}_{r_\sigma}$ denote the combination of these losses. We connect these two sub-networks to the whole architecture using GRLs. GRLs behave as the identity function during the forward pass and invert the gradient sign during the backward pass, hence pushing the parameters to maximize the output loss. During training, this architecture constrains $\Pi$ to produce features in $\mathcal{T}$ and $\mathcal{S}$ with the least information shared between them. Consequently, the update of $\theta_\Pi$ follows $+\nabla_{\theta_\Pi}\mathcal{L}_{info}$.

This constraint efficiently avoids information redundancy between $\mathcal{T}$ and $\mathcal{S}$. However, it does not avoid all the information being pushed into $\mathcal{T}$. Preventing this undesirable behavior is the purpose of condition C4. To that end, we use a cyclic reconstruction scheme. Consider two elements $x$ and $x'$ from $\mathcal{X}$, and their associated $(\tau, \sigma) = \Pi(x)$ and $(\tau', \sigma') = \Pi(x')$. Let $\tilde{x} = \bar{\Pi}(\tau, \sigma')$ be the reconstruction of $\tau$ that uses the style $\sigma'$ of $x'$. A correct allotment of the information between $\mathcal{T}$ and $\mathcal{S}$ requires that the task and style information be preserved in $(\tilde{\tau}, \tilde{\sigma}) = \Pi(\tilde{x})$. So, we wish to have $\tilde{\tau}$ as close as possible to $\tau$, or, alternatively, to have $c(\tilde{\tau})$ as close as possible to $c(\tau)$. Similarly, we wish to have $\tilde{\sigma}$ as close as possible to $\sigma'$. To enforce C4 and avoid the degenerate case where the encoder predicts a constant style for all samples, we force $\tilde{\sigma}$ to lie sufficiently far from $\sigma$ to avoid style confusion. We achieve this with a triplet loss (Schroff et al., 2015) using $\tilde{\sigma}$ as the anchor, $\sigma'$ and $\sigma$ as, respectively, the positive and negative inputs, and a margin $m$. Thus C4 results in minimizing the cyclic reconstruction loss $\mathcal{L}_{cyclic} = \|\tilde{\tau} - \tau\|_2 + max\{\|\tilde{\sigma} - \sigma'\|_2 - \|\tilde{\sigma} - \sigma\|_2 + m, 0\}$.

Overall, the gradient-based update procedure of the network parameters boils down to:

$$\theta_\Pi \leftarrow \theta_\Pi - \alpha\nabla_{\theta_\Pi}(\mathcal{L}_{task} + \mathcal{L}_{reco} - \mathcal{L}_{info} + \mathcal{L}_{cyclic}),$$
$$\theta_{\bar{\Pi}} \leftarrow \theta_{\bar{\Pi}} - \alpha\nabla_{\theta_{\bar{\Pi}}}(\mathcal{L}_{reco} + \mathcal{L}_{cyclic}), \qquad\qquad \theta_c \leftarrow \theta_c - \alpha\nabla_{\theta_c}\mathcal{L}_{task},$$
$$\theta_{r_\tau} \leftarrow \theta_{r_\tau} - \alpha\nabla_{\theta_{r_\tau}}\mathcal{L}_{r_\tau}, \qquad\qquad\qquad\qquad \theta_{r_\sigma} \leftarrow \theta_{r_\sigma} - \alpha\nabla_{\theta_{r_\sigma}}\mathcal{L}_{r_\sigma}.$$

We call this method DiCyR for Disentangled Cyclic Reconstruction.

## 4.2 Task-style disentanglement in the unsupervised domain adaptation case

We propose a variation of DiCyR for UDA, where we replace the decoder $\bar{\Pi}$ by two domain-specific decoders, $\bar{\Pi}_s$ and $\bar{\Pi}_t$. We shall compensate for the lack of labeled data in the target domain by computing cross-domain cyclic reconstructions.

Let $(x_s, y_s)$ be a sample from the source domain and $x_t$ be a sample from the target domain. Let us denote $(\tau_s, \sigma_s) = \Pi(x_s)$ and $(\tau_t, \sigma_t) = \Pi(x_t)$, the corresponding projections in the latent task and style-related information spaces. Then one can define, as in the previous section, $\mathcal{L}_{task}$ as the task-specific loss on the source domain, and $\mathcal{L}_{reco_s}$ and $\mathcal{L}_{reco_t}$ as the reconstruction losses in the source

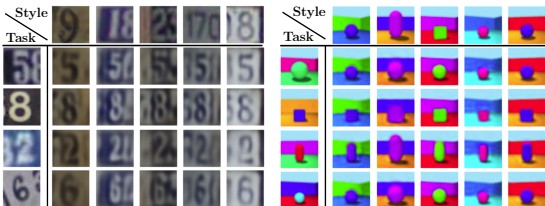

Figure 2: Swapping styles on SVHN and 3D Shapes

and target domains respectively. As previously, we constrain the task-related representation and the style representation not to share information using two networks $r_\tau$ and $r_\sigma$, connected to the main architecture by GRL layers (Figure 1b), allowing the definition of the $\mathcal{L}_{r_\tau}$, $\mathcal{L}_{r_\sigma}$ and $\mathcal{L}_{info}$ losses. Lastly, we exploit cyclic reconstructions in both domains to correctly disentangle the information and hence define the same $\mathcal{L}_{cyclic}$ loss as above.

This disentanglement in the target domain separates the global information in two but does not guarantee that what is being pushed into $\tau$ is really the task-related information. This can only be enforced by cross-domain knowledge (since no correct labels are available in the target domain). Thus, finally, we would like to allow projections from one domain into the other while retaining the task-related information, hence allowing domain adaption. Using the notations above, we construct $x_{ts} = \bar{\Pi}_t(\tau_s, \sigma_t)$, the reconstruction of $x_s$'s task-related information, in the style of $x_t$. This creates an artificial sample in the target domain, whose label is $y_s$. Then, with $(\tau_{ts}, \sigma_{ts}) = \Pi(x_{ts})$, one wishes to have $\tau_{ts}$ match closely $\tau_s$ (or, alternatively, $c(\tau_{ts})$ match closely $y_s$) in order to prevent the loss of task information during the cross-domain projection and thus to constrain the task representations to be domain-invariant. Symmetrically, one can construct the artificial sample $x_{st} = \bar{\Pi}_s(\tau_t, \sigma_s)$ and enforce that $\tau_{st}$ closely matches $\tau_t$. Note that the label of $x_{st}$ is unknown and yet it is still possible to enforce the disentanglement by cyclic reconstruction.

Overall, these terms boil down to a cross-domain cyclic reconstruction loss for UDA $\mathcal{L}_{domain\_cyclic} = \|\tau_s - \tau_{ts}\|_2 + \|\tau_t - \tau_{st}\|_2$.

Finally, the network parameters are updated according to:

$$\theta_\Pi \leftarrow \theta_\Pi - \alpha \nabla_{\theta_\Pi}(\mathcal{L}_{task} + \mathcal{L}_{reco_s} + \mathcal{L}_{reco_t} - \mathcal{L}_{info} + \mathcal{L}_{cyclic} + \mathcal{L}_{domain\_cyclic})$$
$$\theta_{\bar{\Pi}_s} \leftarrow \theta_{\bar{\Pi}_s} - \alpha \nabla_{\theta_{\bar{\Pi}_s}}(\mathcal{L}_{reco_s} + \mathcal{L}_{cyclic}), \qquad \theta_{\bar{\Pi}_t} \leftarrow \theta_{\bar{\Pi}_t} - \alpha \nabla_{\theta_{\bar{\Pi}_t}}(\mathcal{L}_{reco_t} + \mathcal{L}_{cyclic}),$$
$$\theta_c \leftarrow \theta_c - \alpha \nabla_{\theta_c}\mathcal{L}_{task},$$
$$\theta_{r_\tau} \leftarrow \theta_{r_\tau} - \alpha \nabla_{\theta_{r_\tau}}\mathcal{L}_{r_\tau}, \qquad\qquad\qquad \theta_{r_\sigma} \leftarrow \theta_{r_\sigma} - \alpha \nabla_{\theta_{r_\sigma}}\mathcal{L}_{r_\sigma}.$$

## 5 EXPERIMENTAL RESULTS AND DISCUSSION

We first evaluate DiCyR's ability to disentangle the task-related information from the style information in the supervised context. Then we demonstrate DiCyR's efficiency on UDA.[1]

### 5.1 SUPERVISED DISENTANGLEMENT

We evaluate the disentanglement performance of DiCyR by following the protocol introduced by Mathieu et al. (2016). Since we do not use generative models, we only focus on their two first items: *swapping* and *retrieval*. We evaluate DiCyR on the SVHN (Netzer et al., 2011), and 3D Shapes (Burgess & Kim, 2018) disentanglement benchmarks. The task is predicting the central digit in the image for the SVHN dataset, and the shape of the object in the scene for the 3D Shapes dataset.

*Swapping* involves swapping styles between samples and visually assessing the realism of the generated image. It combines the task-related information $\tau_i$ of a sample $x_i$ with the style $\sigma_j$ of another sample $x_j$. We use the decoder to produce an output $\tilde{x}_{ij}$. Figure 2 shows randomly generated outputs on the two datasets. DiCyR produces visually realistic artificial images with the desired styles.

---

[1]Code and pre-trained networks available at `https://github.com/AnonymousDiCyR/DiCyR`.

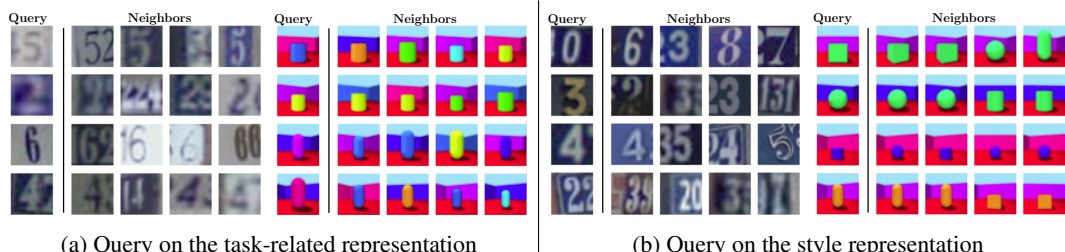

| (a) Query on the task-related representation | (b) Query on the style representation |

Figure 3: Nearest neighbors according to each representation

| Method | SVHN | 3D Shape | floor hue | wall hue | object hue | scale | orientation |
|---|---|---|---|---|---|---|---|
| Full features | 0.98 | 1 | 0.94 | 0.94 | 0.89 | 0.6 | 0.5 |
| Task-related features | 0.98 | 1 | 0.11 | 0.12 | 0.13 | 0.15 | 0.10 |
| Style features only | 0.17 | 0.26 | 0.89 | 0.95 | 0.88 | 0.59 | 0.42 |
| Random guess | 0.10 | 0.25 | 0.10 | 0.10 | 0.10 | 0.125 | 0.067 |

Table 1: Accuracies of a classifier trained to predict factors of style variation on 3D shapes

*Retrieval* concerns finding, in the dataset, the nearest neighbors in the embedding space for an image query. We carry out this search for nearest neighbors using the Euclidean distance on both the task-related and the style representations. A good indicator of the effectiveness of the information disentanglement would be to observe neighbors with the same labels as the query when computing distances on the task-related information space, and neighbors with similar style when using the style information. Figure 3 demonstrate that the neighbors found when using the task-related information are samples with the same label as the query's label and that the neighbors found using the style representation share many characteristics with the query but not necessarily the same labels.

We ran a quantitative evaluation of disentanglement by training a neural network classifier with a single hidden layer of 32 units to predict labels, using either the task-related information alone, or the style information alone. If the information is correctly separated, we expect the classifier trained with task-related information only to get similar performance to a classifier trained with full information. Conversely, the classifier trained with the style information only should reach similar performance to a random guess (10% accuracy on SVHN, 25% on 3D Shapes). Table 1 reports the obtained testing accuracies. It appears that the task-related representation contains enough information to correctly predict labels. We also observe that full disentanglement is closely but not perfectly achieved as the classifier trained only with style information behaves slightly better that random choice. To quantify how much style information is being unduly encoded in the task-related representation, we ran a similar experiment to predict the five other style variation factors in 3D Shapes (floor hue, wall hue, object hue, scale and orientation). The trained classifier reaches accuracies that are very close to a random guess, thus validating the disentanglement quality.

## 5.2 Unsupervised domain adaptation problem

We evaluate DiCyR by performing domain adaptation between the MNIST (LeCun et al., 1998), SVHN, and USPS (Hull, 1994) datasets, and between the Syn-Signs (Ganin & Lempitsky, 2015) and the GTSRB (Stallkamp et al., 2011) datasets. Following common practice in the literature, we trained our network on five different settings: MNIST→USPS, USPS→MNIST, SVHN→MNIST, MNIST→SVHN, and Syn-Signs→GTSRB. We measure the classification performance in the target domain and compare it with state-of-the-art methods (Table 2). We also compare with a baseline classifier that is only trained on the source domain data. Values reported in Table 2 are quoted from their original papers[2]. Our method, without extensive hyperparameter tuning, appears to be on par with the best state-of-the-art methods. DiCyR also outperforms others disentangle-

---

[2]Comparisons might be inexact due to reproducibility concerns (Pineau et al., 2020) and these figures mostly indicate which are the top competing methods.

ment and image-to-image methods. Specifically, DiCyR is only slightly outmatched by DWT and SEDA on the MNIST↔USPS and by SEDA and SHOT in the SVHN→MNIST benchmarks. The MNIST→SVHN case is a particularly challenging benchmark since MNIST images are greyscale and the adaptation to SVHN requires adapting to color images. SEDA makes extensive use of data augmentation to tackle this challenge and is thus the only method displaying convincing results. This hints to a possible enhancement of DiCyR in order to improve its performance. Finally, by introducing a new variation on batch normalization, DWT's contribution is orthogonal to ours and both could be combined. We emphasize that beyond these enhancements, a major advantage of DiCyR lies in the ability to disentangle the information in the target domain without direct supervision.

DiCyR uses GRLs to ensure that no information is shared between $\mathcal{T}$ and $\mathcal{S}$. Similarly to most methods in Table 2, GRLs induce an adversarial optimization problem which is known to yield instability and variance in the resolution performance. In our case, this induces several distinct modes in the distribution of accuracies. It is interesting to note that the majority mode (that of the median) on SVHN→MNIST matches the best known performance. For this reason, we report both the mean and the median on this specific experiment. One might object that condition C3 was expressed in terms of mutual information. Thus, DiCyR only indirectly implements this condition using GRLs. An alternative could be to use an estimator of the mutual information, such as proposed by Belghazi et al. (2018), to directly minimize it (and thus avoid the adversarial setting altogether). Such an approach was explored in the work of Sanchez et al. (2019) to disentangle representations between pairs of images, and would constitute a promising perspective of research for DA.

| Method | Source Target | MNIST USPS | USPS MNIST | SVHN MNIST | MNIST SVHN | Syn-Signs GTSRB |
|---|---|---|---|---|---|---|
| Baseline | | 78.1 | 58.0 | 60.2 | 20.0 | 79.0 |
| DANN (Ganin et al., 2016) | | 85.1 | 73.0 | 73.9 | 35.7 | 88.6 |
| ADDA (Tzeng et al., 2017) | | 89.4 | 90.1 | 76.0 | - | - |
| DSN (Bousmalis et al., 2016) | | 91.3 | - | 82.7 | - | 93.1 |
| DRCN (Ghifary et al., 2016) | | 91.8 | 73.7 | 82.0 | 40.1 | - |
| DiDA (Cao et al., 2018) | | 92.5 | - | 83.6 | - | - |
| SBADA-GAN (Russo et al., 2018) | | 97.6 | 95.0 | 76.1 | 61.1 | 96.7 |
| CyCADA (Hoffman et al., 2018) | | 95.6 | 96.5 | 90.4 | - | - |
| DWT (Roy et al., 2019) | | 99.1 | 98.8 | 97.7 | 28.9 | - |
| SEDA (French et al., 2018) | | 98.2 | 99.5 | 99.3 | 97.0 | 99.3 |
| ACAL (Hosseini-Asl et al., 2019) | | 98.3 | 97.2 | 96.5 | 60.8 | - |
| SHOT (Liang et al., 2020) | | 98.4 | 98.0 | 98.9 | - | - |
| DiCyR (ours) | | 98.4 | 98.3 | 98.5[1] | 23.8 | 97.4 |

[1] median accuracy reported (average accuracy: 95.7 full results distribution are reported in Appendix F).

Table 2: Target domain accuracy, reported as percentages

As in Section 5.1, we evaluate qualitatively the effectiveness of disentanglement, especially in the target domain, and produce visualizations of cross-domain style and task swapping. Here, we combine one domain's task information with the other domain's styles to reconstruct the images of Figures 4a, 4b, 4d, and 4e. The most important finding is that the style information was correctly disentangled from the task-related information in the target domain without the use of any label. Specifically, the rows in these figures show that the class information is preserved when a new style is applied, while the columns illustrate the efficient style transfer allowed by disentanglement.

A desirable property of the task-related encoding is its domain invariance. To evaluate this aspect, we built a t-SNE representation (Hinton & Roweis, 2003) of the task-related features, in order to verify their alignment between domains. Figures 4c and 4f demonstrate this property.

The previous experiments illustrated the use of DiCyR in the context of image classification. The method is, however, quite generic and can be applied in many more contexts. Figure 5 reports the improvement due to applying DiCyR for domain adaptation between the GTA5 (Richter et al., 2016) and the Cityscapes (Cordts et al., 2016) segmentation problems (detailed results in Appendix G).

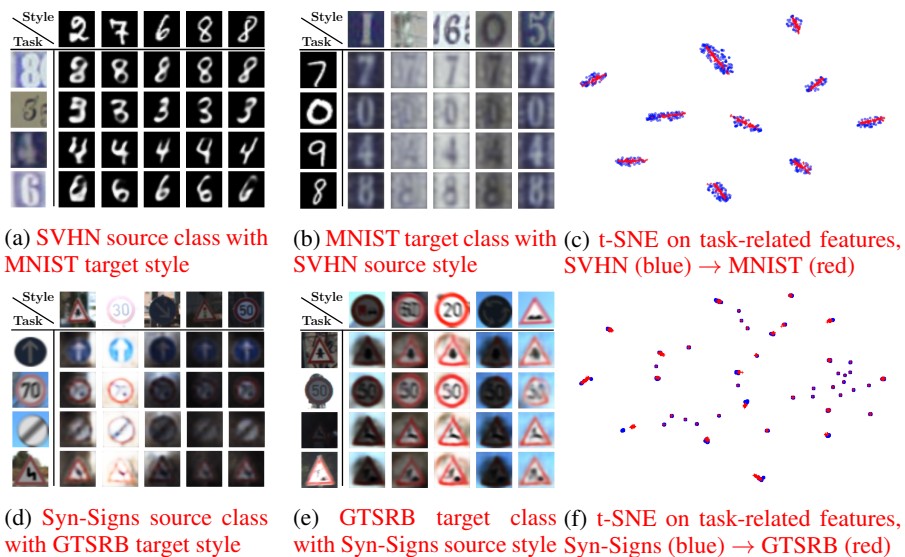

(a) SVHN source class with MNIST target style

(b) MNIST target class with SVHN source style

(c) t-SNE on task-related features, SVHN (blue) → MNIST (red)

(d) Syn-Signs source class with GTSRB target style

(e) GTSRB target class with Syn-Signs source style

(f) t-SNE on task-related features, Syn-Signs (blue) → GTSRB (red)

Figure 4: Cross-domain swapping and feature alignment visualization

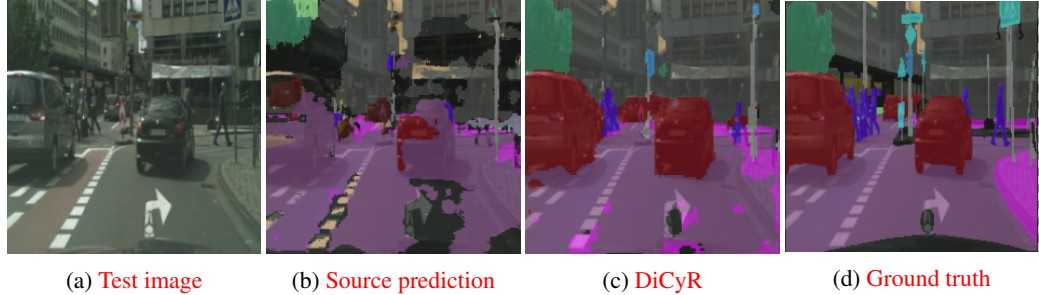

(a) Test image      (b) Source prediction      (c) DiCyR      (d) Ground truth

Figure 5: GTA5 to Cityscapes segmentation

Finally, directly computing the distances on the task-related features in $\mathcal{L}_{domain\_cyclic}$ often leads to unstable results. As hinted in Section 4, using instead a task oriented loss $\mathcal{L}_{domain\_cyclic} = \|c(\tau_s) - y\|_2 + \|c(\tau_t) - c(\tau_{st})\|_2$ stabilizes training and improves the target domain accuracy. Training $c$ with cross-domain projections from the source domain and the corresponding labels improves its generalization to the target domain and forces the encoder to produce task-related features common to both domains. To illustrate this property, consider the following example. In one domain, the digit "7" is written with a middle bar, while in the other it has none. This domain-specific middle bar feature should not be expressed in $\mathcal{T}$; it should be considered as a task-orthogonal style feature. Thus using $c$'s predictions within the domain cyclic loss, instead of distances in $\mathcal{T}$, prevents the encoder from representing the domain-specific features in $\mathcal{T}$ and encourages their embedding in $\mathcal{S}$.

## 6 CONCLUSION

In this work, we introduced a new disentanglement method, called DiCyR, to separate task-related and task-orthogonal style information into different representations in the context of unsupervised domain adaptation. This method also provides a simple and efficient way to obtain disentangled representations for supervised learning problems. Its main features are its overall simplicity, the use of intra-domain and cross-domain cyclic reconstruction, and information separation through Gradient Reversal Layers. The design of this method stems from a formal definition of disentanglement for domain adaptation which, to the best of our knowledge, is new. The empirical evaluation shows that DiCyR performs as well as state-of-the-art methods, while offering the additional benefit of disentanglement, including in the target domains where no label information is available.

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

## A    CROSS-DOMAIN DISENTANGLEMENT VISUALIZATIONS

We report extra cross-domain visualizations similar to those of Section 5.2 in Figures 6 and 7.

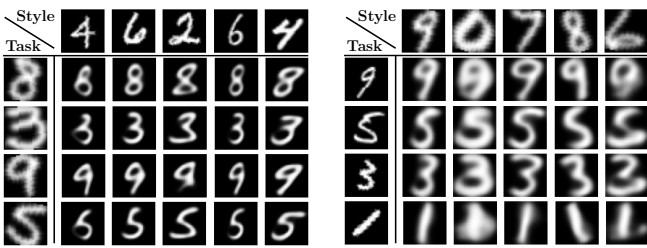

(a) Combination of source class with target style

(b) Combination of target class with source style

Figure 6: Cross-domain swapping, USPS→MNIST

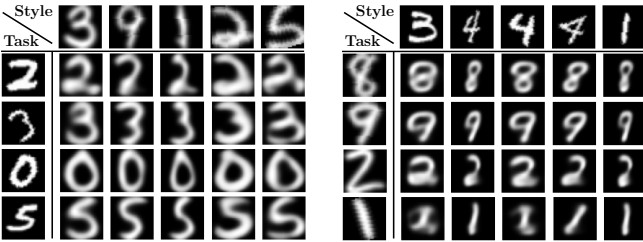

(a) Combination of source class with target style

(b) Combination of target class with source style

Figure 7: Cross-domain swapping, MNIST→USPS

## B    NETWORK ARCHITECTURE AND TRAINING HYPER-PARAMETERS

The paragraphs below detail the network architctures used in the experiments of Section 5. It should be noted that neither these architectures, nor the associated hyper-parameters have been extensively and finely tuned to their respective tasks, as the goal of this contribution was to provide a generic, robust method. Thus it is likely that performance gains can still be obtained on this front.

### B.1    SINGLE DOMAIN SUPERVISED DISENTANGLEMENT EXPERIMENTS

This section describes the network architecture and the hyper-parameters used in the experiments of Section 5.1. The encoder $\Pi$ is composed of shared layers (layers 1 to 6 in the table below), followed by the specific task-related and style encodings. Those final layers are denoted $\Pi_\tau$ and $\Pi_\sigma$ in the tables below. For the sake of implementation simplicity, we chose to project samples from the source domain and samples from the target domain into two separate style embeddings (one for each domain). Thus $\Pi_\sigma$ is actually duplicated in two heads $\Pi_{\sigma,s}$ and $\Pi_{\sigma,t}$ with the same structure and output space. We used the exact same network architectures for both the 3D shapes and SVHN datasets, the only difference being the dimension of the embeddings $\mathcal{T}$ and $\mathcal{S}$. In all experiments, we applied a coefficient $\beta_{reco} = 5$ to $\mathcal{L}_{reco}$ and $\beta_{cyclic} = 0.1$ to $\mathcal{L}_{cyclic}$ in the global loss. We also use a $\beta_{info}$ on $\mathcal{L}_{info}$; this coefficient increases linearly from $10^{-2}$ to 10 over the first 10 epochs and remains at 10 afterwards (see Section C for a discussion on this coefficient). Convergence was reached within 50 epochs. We used Adam (Kingma & Ba, 2015) as an optimizer with a learning rate $lr = 5 \cdot 10^{-4}$ for the first 30 epochs and $lr = 5 \cdot 10^{-5}$ for the last 20 epochs. The following tables summarize the architectures of all sub-networks.

| Architecture of $\Pi$, single domain case | | |
|---|---|---|
| Layer | Type | Parameters |
| 1 | Conv2D | filters=32, kernel=$5 \times 5$, stride=1, padding=2, activation=ReLU |
| 2 | Max Pooling | filters=$2 \times 2$, stride=2 |
| 3 | Conv2D | filters=64 for 3DShapes or 32 for SVHN |
| | | kernel=$5 \times 5$, stride=1, padding=2, activation=ReLU |
| 4 | Max Pooling | filters=$2 \times 2$, stride=2 |
| 5 | Conv2D | filters=128 for 3DShapes or 64 for SVHN |
| | | kernel=$3 \times 3$, stride=1, padding=1, activation=ReLU |
| 6 | Dense | nb_neurons=1024, activation=ReLU |
| $\Pi_\tau$ | Dense | nb_neurons=20 for 3D shapes or 150 for SVHN, activation=ReLU |
| $\Pi_\sigma$ | Dense | nb_neurons=20 for 3D shapes or 150 for SVHN, activation=ReLU |

| Architecture of $\bar{\Pi}$, single domain case | | |
|---|---|---|
| Layer | Type | Parameters |
| 1 | Dense | nb_neurons=1024, activation=ReLU |
| 2 | Dense | nb_neurons=64, activation=ReLU |
| 3 | Conv2D | filters=64, kernel=$3 \times 3$, stride=1, padding=1, activation=ReLU |
| 4 | Upsample | scale_factor=2 |
| 5 | Conv2D | filters=32, kernel=$5 \times 5$, stride=1, padding=2, activation=ReLU |
| 6 | Upsample | scale_factor=2 |
| 7 | Conv2D | filters=3, kernel=$5 \times 5$, stride=1, padding=2, activation=Sigmoid |

| Architecture of $c$, single domain case | | |
|---|---|---|
| Layer | Type | Parameters |
| 1 | Dropout | p=0.2 for 3DShapes and 0.55 for SVHN |
| 2 | Dense | nb_neurons=nb_labels, activation=Softmax |

| Architecture of $r_\tau$ and $r_\sigma$, 3DShapes single domain case | | |
|---|---|---|
| Layer | Type | Parameters |
| 1 | Gradient Reversal Layer | |
| 2 | Dense | nb_neurons=100, activation=ReLU |
| 3 | Dense | nb_neurons=20, activation=Linear |

| Architecture of $r_\tau$ and $r_\sigma$, SVHN single domain case | | |
|---|---|---|
| Layer | Type | Parameters |
| 1 | Gradient Reversal Layer | |
| 2 | Dense | nb_neurons=100, activation=ReLU |
| 3 | Dense | nb_neurons=100, activation=ReLU |
| 4 | Dense | nb_neurons=150, activation=Linear |

## B.2 UNSUPERVISED DOMAIN ADAPTATION EXPERIMENTS

This section describes the network architecture and the hyper-parameters used in the experiments of Section 5.2. The encoder $\Pi$ is composed of shared layers (layers 1 to 9 in the tables below), followed by the specific task-related and style encodings. Those final layers are denoted $\Pi_\tau$ and $\Pi_\sigma$ in the tables below. For the sake of implementation simplicity, we chose to project samples from the source domain and samples from the target domain into two separate style embeddings (one for each domain). Thus $\Pi_\sigma$ is actually duplicated in two heads $\Pi_{\sigma,s}$ and $\Pi_{\sigma,t}$ with the same structure and output space. In all experiments, in the global loss, we applied a coefficient $\beta_{cyclic} = 0.1$ to $\mathcal{L}_{cyclic}$ and $\beta_{domain\_cyclic}$ to $\mathcal{L}_{domain\_cyclic}$, with $\beta_{domain\_cyclic}$ increasing linearly from 0 to 10 during the 10 first epochs and remaining at 10 afterwards (see Section C for a discussion on this coefficient). Convergence was reached within 50 epochs (generally within 30 epochs). We used Adam (Kingma & Ba, 2015) as an optimizer with a learning rate $lr = 5 \cdot 10^{-4}$ for the first 30 epochs and $lr = 5 \cdot 10^{-5}$ for the last 20 epochs. The following tables summarize the architectures of all sub-networks.

| Architecture of $\Pi$, SVHN$\leftrightarrow$MNIST case | | |
|---|---|---|
| Layer | Type | Parameters |
| 1 | Conv2D | filters=32, kernel=5 × 5, stride=1, padding=2, activation=Linear Instance Normalization |
| 2 | Max Pooling | filters=2 × 2, stride=2 |
| 3 | Conv2D | filters=32 for SVHN$\rightarrow$MNIST 64 for MNIST$\rightarrow$SVHN kernel=5 × 5, stride=1, padding=2, activation=Linear Instance Normalization |
| 4 | Instance Normalization | |
| 5 | Max Pooling | filters=2 × 2, stride=2 |
| 6 | Conv2D | filters=32 for SVHN$\rightarrow$MNIST 128 for MNIST$\rightarrow$SVHN kernel=3 × 3, stride=1, padding=1, activation=Linear Instance Normalization |
| 7 | Dense | nb_neurons=1024, activation=ReLU |
| $\Pi_\tau$ | Dense | nb_neurons=75 for SVHN$\rightarrow$MNIST activation=ReLU nb_neurons=200 for MNIST$\rightarrow$SVHN, activation=ReLU |
| $\Pi_\sigma$ | Dense | nb_neurons=75 for SVHN$\rightarrow$MNIST activation=ReLU nb_neurons=200 for MNIST$\rightarrow$SVHN, activation=ReLU |

| Architecture of $\bar\Pi_s$ and $\bar\Pi_t$, SVHN$\leftrightarrow$MNIST case | | |
|---|---|---|
| Layer | Type | Parameters |
| 1 | Dense | nb_neurons=1024, activation=ReLU |
| 2 | Dense | nb_neurons=64, activation=ReLU |
| 3 | Conv2D | filters=32 for SVHN$\rightarrow$MNIST 64 for MNIST$\rightarrow$SVHN kernel=3 × 3, stride=1, padding=1, activation=ReLU |
| 4 | Upsample | scale_factor=2 |
| 5 | Conv2D | filters=32, kernel=5 × 5, stride=1, padding=2, activation=ReLU |
| 6 | Upsample | scale_factor=2 |
| 7 | Conv2D | filters=3, kernel=5 × 5, stride=1, padding=2, activation=Sigmoid |

| Architecture of $c$, SVHN$\leftrightarrow$MNIST case | | |
|---|---|---|
| Layer | Type | Parameters |
| 1 | Dropout | p=0.55 |
| 2 | Dense | nb_neurons=10, activation=Softmax |

| Architecture of $r_\tau$ and $r_\sigma$, SVHN$\leftrightarrow$MNIST case | | |
|---|---|---|
| Layer | Type | Parameters |
| 1 | Gradient Reversal Layer | |
| 2 | Dense | nb_neurons=100, activation=ReLU |
| 3 | Dense | nb_neurons=75 for SVHN$\rightarrow$MNIST activation=Linear nb_neurons=200 for MNIST$\rightarrow$SVHN, activation=Linear |

| Architecture of $\Pi$, MNIST$\leftrightarrow$USPS case | | |
|---|---|---|
| Layer | Type | Parameters |
| 1 | Conv2D | filters=50, kernel=5 × 5, stride=1, padding=2, activation=ReLU, Batch Norm. |
| 2 | Max Pooling | filters=2 × 2, stride=2 |
| 3 | Conv2D | filters=75, kernel=5 × 5, stride=1, padding=2, activation=ReLU, Batch Norm. |
| 4 | Max Pooling | filters=2 × 2, stride=2 |
| 5 | Conv2D | filters=100, kernel=3 × 3, stride=1, padding=1, activation=ReLU, Batch Norm. |
| 6 | Dense | nb_neurons=1024, activation=ReLU |
| $\Pi_\tau$ | Dense | nb_neurons=150, activation=ReLU |
| $\Pi_\sigma$ | Dense | nb_neurons=150, activation=ReLU |

| Architecture of $\bar{\bar{\Pi}}_s$ and $\bar{\bar{\Pi}}_t$, MNIST↔USPS case | | |
|---|---|---|
| Layer | Type | Parameters |
| 1 | Dense | nb_neurons=1024, activation=ReLU |
| 2 | Dense | nb_neurons=64, activation=ReLU |
| 3 | Conv2D | filters=100, kernel=$3 \times 3$, stride=1, padding=1, activation=ReLU |
| 4 | Upsample | scale_factor=2 |
| 5 | Conv2D | filters=50, kernel=$5 \times 5$, stride=1, padding=2, activation=ReLU |
| 6 | Upsample | scale_factor=2 |
| 7 | Conv2D | filters=1, kernel=$5 \times 5$, stride=1, padding=2, activation=Sigmoid |

| Architecture of $c$, MNIST↔USPS case | | |
|---|---|---|
| Layer | Type | Parameters |
| Layer | Type | Parameters |
| 1 | Dropout | p=0.55 |
| 2 | Dense | nb_neurons=10, activation=Softmax |

| Architecture of $r_\tau$ and $r_\sigma$, MNIST↔USPS case | | |
|---|---|---|
| Layer | Type | Parameters |
| 1 | Gradient Reversal Layer | |
| 2 | Dense | nb_neurons=100, activation=ReLU |
| 3 | Dense | nb_neurons=150, activation=Linear |

| Architecture of $\Pi$, Syn-Signs↔GTSRB case | | |
|---|---|---|
| Layer | Type | Parameters |
| 1 | Conv2D | filters=32, kernel=$5 \times 5$, stride=1, padding=2, activation=ReLU, Batch Norm. |
| 2 | Max Pooling | filters=$2 \times 2$, stride=2 |
| 3 | Conv2D | filters=64, kernel=$5 \times 5$, stride=1, padding=2, activation=ReLU, Batch Norm. |
| 4 | Max Pooling | filters=$2 \times 2$, stride=2 |
| 5 | Conv2D | filters=128, kernel=$3 \times 3$, stride=1, padding=1, activation=ReLU, Batch Norm. |
| 6 | Max Pooling | filters=$2 \times 2$, stride=2 |
| 7 | Conv2D | filters=128, kernel=$3 \times 3$, stride=1, padding=1, activation=ReLU, Batch Norm. |
| 8 | Dense | nb_neurons=1024, activation=ReLU |
| $\Pi_\tau$ | Dense | nb_neurons=75, activation=ReLU |
| $\Pi_\sigma$ | Dense | nb_neurons=75, activation=ReLU |

| Architecture of $\bar{\bar{\Pi}}_s$ and $\bar{\bar{\Pi}}_t$, Syn-Signs↔GTSRB case | | |
|---|---|---|
| Layer | Type | Parameters |
| 1 | Conv2D | filters=64, kernel=$4 \times 4$, stride=1, padding=1, activation=ReLU |
| 2 | Upsample | scale_factor=2 |
| 3 | Conv2D | filters=32, kernel=$3 \times 3$, stride=1, padding=2, activation=ReLU |
| 4 | Upsample | scale_factor=2 |
| 5 | Conv2D | filters=32, kernel=$3 \times 3$, stride=1, padding=2, activation=ReLU |
| 6 | Upsample | scale_factor=2 |
| 7 | Conv2D | filters=1, kernel=$3 \times 3$, stride=1, padding=2, activation=Sigmoid |

| Architecture of $c$ Syn-Signs↔GTSRB | | |
|---|---|---|
| Layer | Type | Parameters |
| 1 | Dropout | p=0.55 |
| 2 | Dense | nb_neurons=10, activation=Softmax |

| Architecture of $r_\tau$ and $r_\sigma$, Syn-Signs↔GTSRB case | | |
|---|---|---|
| Layer | Type | Parameters |
| 1 | Gradient Reversal Layer | |
| 2 | Dense | nb_neurons=100, activation=ReLU |
| 3 | Dense | nb_neurons=150, activation=Linear |

For the GTA5 $\rightarrow$ Cityscapes experiment, we based our network's architecture on the one proposed by (Romera et al., 2017).

| \multicolumn{2}{c}{Architecture of $\Pi$ GTA5 $\rightarrow$ Cityscapes case} | |
|---|---|
| Layer | Type |
| 1 | Downsampler block, filters=16 |
| 2 | Downsampler block, filters=64 |
| 3-7 | Non-bt-1D. |
| 8 | Downsampler block, filters=128 |
| 9 | Non-bt-1D (dilated 2) |
| 10 | Non-bt-1D (dilated 2) |
| 11 | Non-bt-1D (dilated 2) |
| 12 | Non-bt-1D (dilated 2) |
| 13 | Non-bt-1D (dilated 2) |
| 14 | Non-bt-1D (dilated 2) |
| 15 | Non-bt-1D (dilated 2) |
| 16 | Non-bt-1D (dilated 2) |
| $\Pi_\tau$ | Conv2d filter=114, kernel=1, activation=ReLU |
| $\Pi_\sigma$ | Conv2d filter=14, kernel=1, activation=ReLU |

| \multicolumn{2}{c}{Architecture of $c$, $\bar{\bar{\Pi}}_s$ and $\bar{\bar{\Pi}}_t$ GTA5 $\rightarrow$ Cityscapes case} | |
|---|---|
| Layer | Type |
| 1 | Conv2D filters=64 kernel=3 $\times$ 3, stride=2, padding=1, activation=ReLU |
| 2 | Upsample scale_factor=2 |
| 3-4 | non-bottleneck-1D |
| 5 | Conv2D filters=16 kernel=3 $\times$ 3, stride=2, padding=1, activation=ReLU |
| 6 | Upsample scale_factor=2 |
| 7-8 | non-bottleneck-1D |
| 9 | Upsample scale_factor=2 |
| 10 | Conv2D filters=20 for $c$ and filters=3 for $\bar{\bar{\Pi}}_s$ and $\bar{\bar{\Pi}}_t$ kernel=3 $\times$ 3, stride=2, padding=1, activation=ReLU |

## C  DISCUSSION ON THE $\beta_{info}$ AND $\beta_{domain\_cyclic}$ COEFFICIENTS SCHEDULE

Although the schedule on $\beta_{info}$ (single domain case) and $\beta_{domain\_cyclic}$ (domain adaptation case) is not absolutely necessary, we found out it helped the overall convergence. These coefficients gradually increase the weight of the information disentanglement objective and the cross-domain reconstruction objective. This assigns more importance to learning a good predictor $c \circ \Pi$ during early stages. From this perspective, gradually increasing $\beta_{info}$ can be seen as gradually removing task-useless information from $\mathcal{T}$ and transferring it to $\mathcal{S}$. Similarly, increasing $\beta_{domain\_cyclic}$ corresponds to letting the network discover disentangled representations before aligning them across domains.

As previously mentioned, our goal in this study was to provide a robust disentanglement method that permits domain adaptation. Therefore, no complete hyper-parameter study and tuning was performed and these findings are thus reported as such and might be incomplete. Refining the understanding of the influence of the different $\beta$ coefficients is closer to the problem of meta-learning and is beyond the scope of this paper.

## D  INFLUENCE OF BATCH NORMALIZATION AND DROPOUT ON DICYR

Batch normalization (Ioffe & Szegedy, 2015) is an efficient way to reduce the discrepancy between the source and target distributions statistics. We noticed that, for the specific SVHN $\rightarrow$ MNIST setting, using instance normalization (Ulyanov et al., 2016) slightly improves the target domain accuracy. Normalizing across chanels, the instance normalization layers helps the networks to be agnostic to the image contrast which is particularly strong in MNIST. We also noticed that using a large dropout in the sub-network $c$, and small embedding dimensions for $\Pi$'s ouputs improves

both the disentanglement quality and the target domain accuracy. We conjecture that the information bottleneck induced forces the task-related representation to be as concise as possible and thus encourages disentanglement.

# E    COMPLIMENTARY INFORMATION RELATED TO THE MACHINE LEARNING REPRODUCIBILITY CHECKLIST

All the experiments from section 5 were run on a Google Cloud Platform n1-standard-8 virtual machine (8 virtual cores, 30Go RAM, Nvidia P100 GPU) except the experiment on GTA5 → Cityscapes for which two Nvidia P100 GPU were used. The code corresponding to the experiments, a list of dependencies, and pre-trained models are available at `https://github.com/AnonymousDiCyR/DiCyR` Details about each experiments are reported on Table 3.

| Experiment | Image size | Batch size | Number of epochs | Time by epoch | Number of experiment repetitions |
|---|---|---|---|---|---|
| 5.1 SVHN | 32×32 | 64 | 50 | 35 s | 50 |
| 5.1 3D Shapes | 32×32 | 64 | 50 | 15 s | 5 |
| 5.2 MNIST→USPS | 32×32 | 128 | 150 | 11 s | 20 |
| 5.2 USPS→MNIST | 32×32 | 128 | 150 | 11 s | 20 |
| 5.2 MNIST→SVHN | 32×32 | 128 | 50 | 40 s | 50 |
| 5.2 SVHN→MNIST | 32×32 | 64 | 50 | 40 s | 50 |
| 5.2 Syn-Signs→GTSRB | 64×64 | 64 | 150 | 65 s | 10 |
| 5.2 GTA5→Cityscapes | 256 × 256 | 100[1] | 200 | 128 s | 3 |

[1] To account for GPU memory limits, the gradients were accumulated over 5 batches of size 20.

Table 3: Experimental setup

# F    DISTRIBUTION OF RESULTS

In Figure 8 we report the distribution of testing accuracies for the SVHN→MNIST benchmark of Section 5.2. Most of the accuracies are distributed within three modes with low variance, centered respectively around 89%, 94% and 98%, the latter being also the median and the majority mode. This sheds a more detailed light on the results reported in Section 5.2. As discussed therein, we conjecture these modes stem from the local equilibria of the adversarial optimization problem induced by the GRLs in DiCyR. Thus we expect that avoiding this problem altogether in the formulation of DiCyR could improve these resulting distributions.

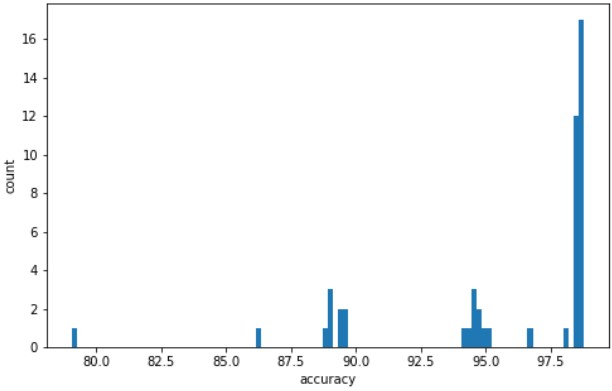

Figure 8: Distribution of testing accuracies across experiments for SVHN→MNIST

## G  DOMAIN ADAPTATION FOR THE GTA5→CITYSCAPES SEGMENTATION TASK

Here we report on the application of DiCyR for the segmentation task in GTA5 (Richter et al., 2016) and Cityscapes (Cordts et al., 2016). GTA5 is the source domain, where the ground truth of image segments is provided, and the goal is to reach efficient segmentation on the Cityscapes dataset.

Figure 5 provided a first visual illustration of the benefits provided by DiCyR. Figure 9 provides yet other such examples. Column 9a presents the testing image from the Cityscapes dataset, column 9b shows the application on the testing image of a classifier trained only on the GTA5 images, column 9c shows the segmentation obtained by DiCyR, which can be compared to the ground truth (column 9d).

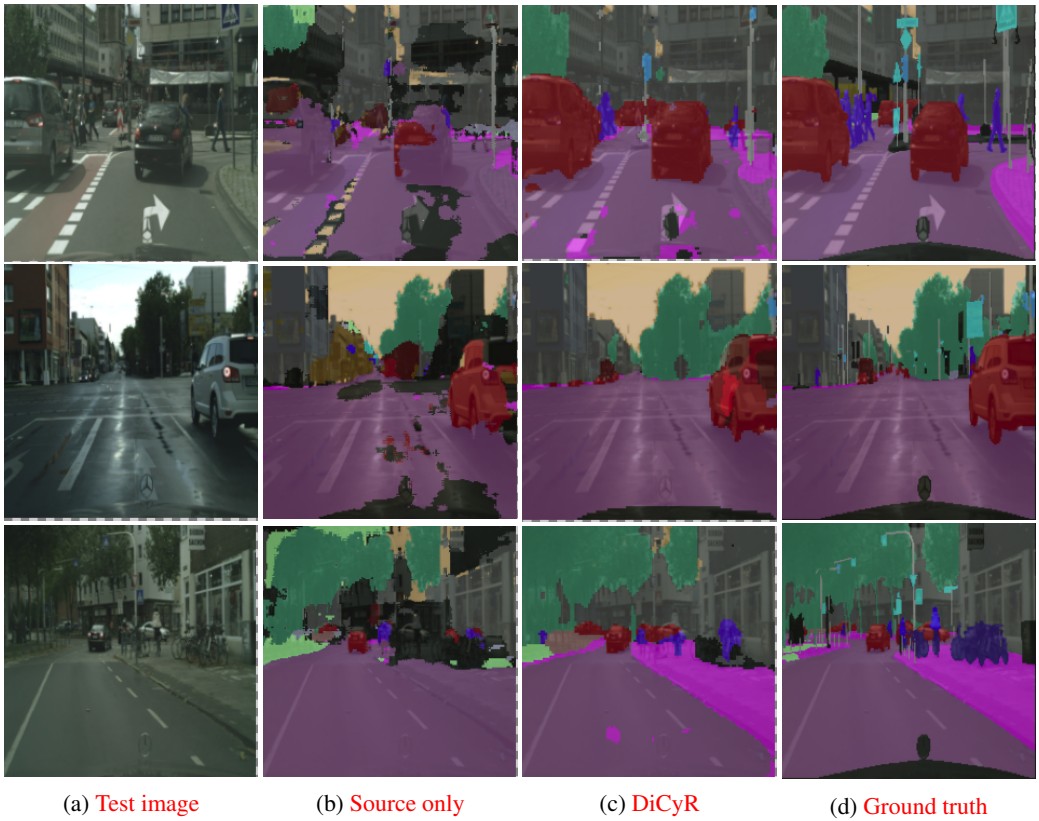

    (a) Test image        (b) Source only        (c) DiCyR        (d) Ground truth

Figure 9: GTA5 to Cityscapes segmentation

Table 4 reports the Intersection over Union criterion (the Jaccard index) for each object class in the Cityscapes images. We compare the seminal approach of Hoffman et al. (2016) coined "FCNs in the wild", which served as a baseline for CyCADA (Hoffman et al., 2018), and the application of DiCyR. These results are preliminary[3] and did not benefit from any hyperparameter or architecture tuning. The purpose of this table is to report the out-of-the-box performance of DiCyR and validate the rationale behind performing disentanglement, on a challenging, large scale problem.

---

[3]They were obtained following a very valuable suggestion by a reviewer.

| | road | sidewalk | building | wall | fence | pole | traffic light | traffic sign | vegetation | terrain | sky | person | rider | car | truck | bus | train | motorcycle | bicycle | **mIoU** |
|---|---|---|---|---|---|---|---|---|---|---|---|---|---|---|---|---|---|---|---|---|
| Source only | 68.6 | 11.8 | 57.4 | 5.4 | 2.5 | 11.3 | 6.6 | 0.9 | 65.5 | 12.9 | 42.1 | 13.1 | 1.7 | 41.9 | 4.7 | 3.8 | 2.8 | 1.8 | 0.0 | 18.7 |
| FCNs in the wild | 70.4 | 32.4 | 62.1 | 14.9 | 5.4 | 10.9 | 14.2 | 2.7 | 79.2 | 21.3 | 64.6 | 44.1 | 4.2 | 70.4 | 8.0 | 7.3 | 0.0 | 3.5 | 0.0 | 27.1 |
| CyCADA | 85.2 | 37.2 | 76.5 | 21.8 | 15.0 | 23.8 | 22.9 | 21.5 | 80.5 | 31.3 | 60.7 | 50.5 | 9.0 | 76.9 | 17.1 | 28.2 | 4.5 | 9.8 | 0.0 | 35.4 |
| DiCyR | 69.2 | 15.5 | 68.2 | 15.4 | 9.2 | 12 | 9.6 | 1.2 | 70.1 | 32.1 | 60.9 | 28.6 | 0.3 | 59.6 | 8.9 | 8.7 | 2.0 | 4.4 | 0.0 | 25.0 |

Table 4: Intersection over Union (IoU) criterion for each object class

