# OpenReview forum: "Disentangled cyclic reconstruction for domain adaptation"
_ICLR.cc/2021/Conference — Reject_

### Official Review · AnonReviewer4 · 2020-10-28
**Interesting idea, but incompetitive performance**

**Rating:** 5
**Confidence:** 3

**Review:**

This paper studies the domain adaptation problem by addressing the challenge of splitting task-specific and task-orthogonal information in the target domain using the proposed disentangled cyclic reconstruction method. The authors further develop a variant for the unsupervised domain adaption (UDA) task. The authors argue that the existing adversarial classifier based UDA solutions do not guarantee that the domain specific information does not contain any information that overlaps with the shared information in the target domain. Another shortcoming of these baselines is the learned representation in their domain-invariant feature space might not allow for accurate labeling in the target domain. To solves these limitations, the authors directly minimize the information sharing between representation, instead of using domain adversarial classifier and adversarial label predictor.

This paper is well structured and easy to follow. The authors make several interesting and insightful statements. While the justifications of these main statements are not well presented theoretically/experimentally.

In page 3, the authors argue that the existing solutions achieve a domain-invariant feature space in which the representation might not allow for accurate labeling in the target domain. The following explanation reads vague to me. A qualitive example or an analytical experiment is encouraged to have to justify the mentioned “some elements” will hinder the performance.

My major concern is the evaluation results compared with the baselines is not competitive. As shown in Table 2, in the benchmark MNIST/USPS/SVHN datasets, the proposed method is inferior to the baselines like SEDA, DWT, SBADA-GAN. Since these datasets are relatively simple, I would encourage the authors to conduct experiments on other more challenging UDA benchmarks, like DomainNet [ref-1] and VisDA2017 [ref-2]
[ref-1] X. Peng, et al., Moment matching for multi-source domain adaptation. In ICCV, 2019.
[ref-2] X . Peng, et al. Visda:  The visual domain adaptation challenge.arXiv preprint arXiv:1710.06924, 2017.

Missing some recent related works, including but not limted to
[ref-3] Ryuhei Takahashi, et al: Partially-Shared Variational Auto-encoders for Unsupervised Domain Adaptation with Target Shift. ECCV 2020
[ref-4] Jian Liang, et al: Do We Really Need to Access the Source Data? Source Hypothesis Transfer for Unsupervised Domain Adaptation, ICML 2020

Other minor point. In page 3, last sentence in paragraph 4, “ a discriminator with a an inverted label loss”. Remove ‘a’.

Updates: Thanks for the authors' response. Some of my queries were clarified. However, unfortunately, I still think more needs to be done to show the superiority of the results. I retain my original decision.

---

> ### Author Response · Authors · 2020-11-14
> **Answer to AnonReviewer4, question 3**
>
> > Missing some recent related works, including but not limted to [ref-3] Ryuhei Takahashi, et al: Partially-Shared Variational Auto-encoders for Unsupervised Domain Adaptation with Target Shift. ECCV 2020 [ref-4] Jian Liang, et al: Do We Really Need to Access the Source Data? Source Hypothesis Transfer for Unsupervised Domain Adaptation, ICML 2020
>
> Thank you for pointing us towards this work. We added it to the related work and to our benchmark in Table 2.

---

> ### Author Response · Authors · 2020-11-14
> **Answer to AnonReviewer4, question 2**
>
> > My major concern is the evaluation results compared with the baselines is not competitive. As shown in Table 2, in the benchmark MNIST/USPS/SVHN datasets, the proposed method is inferior to the baselines like SEDA, DWT, SBADA-GAN. Since these datasets are relatively simple, I would encourage the authors to conduct experiments on other more challenging UDA benchmarks, like DomainNet [ref-1] and VisDA2017 [ref-2] [ref-1] X. Peng, et al., Moment matching for multi-source domain adaptation. In ICCV, 2019. [ref-2] X . Peng, et al. Visda: The visual domain adaptation challenge.arXiv preprint arXiv:1710.06924, 2017.
>
> As mentioned in the experimental section we did not undertake extensive hyperparameters tuning.
> However, it turns out that most of the methods having superior performance make use of data-augmentation.
> We conducted a new series of experiments using the exact same data-augmentation used in DWT and SEDA (affine transformation and gaussian blur).
> This resulted in an improvement in our performances and in the stability of our method allowing DiCyR to overcome SBADA-GAN performances in all benchmarks and to obtain comparable performances to the ones obtained by DWT and SEDA.
> Those improved results are reported in Table 2.
> Moreover, we would like to emphasize the fact that our method obtains the best performance among the methods using disentanglement and image-to-image translation, and that it is orthogonal to SEDA and DWT in the sense that it could be combined with them.
> We also added a new benchmark to the results reported in Table 2, with the Syn-Signs→GTSRB dataset.
> However, we believe that your suggestion to conduct experiments on other more challenging UDA benchmarks is absolutely necessary and relevant. Considering the short amount of time and our limited computing resources, we will do our best to provide an additional experiment on VisDA2017 or Cityscapes-to-GTA5 before the end of the rebuttal period.

---

> > ### Author Response · Authors · 2020-11-23
> > **Requested additional experiments**
> >
> > As suggested, we ran an additional experiment on a large scale dataset: GTA5 -> Cityscapes. We assessed both the model performance and computational times. The results were included in the paper and detailed results were added to the Appendix, sections E and G.

---

> ### Author Response · Authors · 2020-11-14
> **Answer to AnonReviewer4, question 1**
>
> > In page 3, the authors argue that the existing solutions achieve a domain-invariant feature space in which the representation might not allow for accurate labeling in the target domain. The following explanation reads vague to me. A qualitive example or an analytical experiment is encouraged to have to justify the mentioned “some elements” will hinder the performance.
>
> We believe that it is possible to train an encoder to project the elements of both domains into the same regions of a latent space, thus making it impossible for a domain discriminator to predict the original domain of the projected elements without guaranteeing that elements with the same label are projected in the same regions.
> Consider a binary classification problem (classes $c_1$ and $c_2$) and two domains ($d_1$ and $d_2$).
> Let $(c_1,d_1)$ denote samples of class $c_1$ in $d_1$.
> It is possible to construct an encoding that projects $(c1,d1)$ and $(c2,d2)$ to the same feature values.
> The same holds for $(c_1,d_2)$ and $(c_2,d_1)$ for different feature values.
> This encoding allows discriminating between classes in $d_1$.
> It also fools a domain discriminator since it does not allow predicting the original domain of a projected element.
> However, applying the classification function learned on $d_1$ to the projected $d_2$ elements leads to catastrophic predictions.
> We added this example in the mentioned section.
> Do you think that this example helps to clarify our claim?

---

### Official Review · AnonReviewer2 · 2020-10-28
**The paper proposes a disentanglement method, named DiCyR in the problem of unsupervised domain adaptation.**

**Rating:** 6
**Confidence:** 3

**Review:**

#####################

Summary:

The paper proposes a disentanglement method, named DiCyR in the problem of unsupervised domain adaptation.

#####################

Reason for score:

Overall, the paper is above the borderline. I like the idea of utilizing disentanglement learning with a cycle constraint into the unsupervised domain adaptation issue. My major concern is about some unclear parts described in the paper and insufficient experimental comparison (see cons below). Hopefully, it would be grateful that the authors could address my concerns during the rebuttal period.

#####################

Pros:

(1) The idea in the paper to introduce disentanglement learning based on cyclic reconstruction to unsupervised domain adaptation is very novel and interesting, which can split the captured information from data into task-related representation and context-based features to help improve the domain generality of the model.

(2) The disentanglement learning to split the information into a task-related representation and a context representation is reasonable and promising, which is beneficial to the task model adapted in different domains.

(3) Extensive experimental results on several widely-adopted benchmarks show the effectiveness of the proposed model comparing with other state-of-the-art methods.

#####################

Cons:

(1) In the paper, the introduced disentanglement learning method can split information captured from images into a task-specific feature and another task-orthogonal representation. However, the proposed method cannot consider the domain-specific and domain-orthogonal information, which may degrade the performance in experiments and deserves to be discussed further. So how to learn the domain-invariant representation in the model? Please explain more details.

(2) In the part of the experiment in the paper, the authors conduct the experiments on the task of image style transfer. However, the task of semantic segmentation on Cityscapes is always widely adopted in related UDA papers, which is not included in the paper. Please consider adding such experimental results and analysis to demonstrate the effectiveness of the proposed method in the paper if feasible.

(3) In the paper, the author claims the proposed model can achieve high computational efficiency, but there are no figures or tables to demonstrate it. Please consider listing the comparison results of training time and test time in the experimental part.

#####################

Questions during the rebuttal period:

Please address and clarify the cons above. Thank you!

---

> ### Author Response · Authors · 2020-11-14
> **Answer to AnonReviewer2, question 3**
>
> > In the paper, the author claims the proposed model can achieve high computational efficiency, but there are no figures or tables to demonstrate it. Please consider listing the comparison results of training time and test time in the experimental part.
>
> Following your suggestion, we ran an experiment to double-check the convergence time of the state of the art methods (using the publicly available codes) with their reported hyperparameters (on which we had based our previous statement). In the light of these experiments, we nuanced the statement that our method is faster: it seems to be equivalent to most state-of-the-art methods (in particular SEDA, DWT, and SHOT) in number of epochs to convergence. For this reason, we rephrased this statement in the manuscript and would like to thank you for encouraging us to undertake this experiment, which also helped us discover new hyperparameters improving our results.

---

> ### Author Response · Authors · 2020-11-14
> **Answer to AnonReviewer2, question 2**
>
> >In the part of the experiment in the paper, the authors conduct the experiments on the task of image style transfer. However, the task of semantic segmentation on Cityscapes is always widely adopted in related UDA papers, which is not included in the paper. Please consider adding such experimental results and analysis to demonstrate the effectiveness of the proposed method in the paper if feasible.
>
> As mentioned in the answers to AnonReviewer3, we are doing our best to provide an experiment on the Cityscapes dataset before the end of the rebuttal period. This remains a challenging task for us in such a short time due to our limited computational resources.

---

> > ### Author Response · Authors · 2020-11-23
> > **Requested additional experiments**
> >
> > As suggested, we ran an additional experiment on a large scale dataset: GTA5 -> Cityscapes. We assessed both the model performance and computational times. The results were included in the paper and detailed results were added to the Appendix, sections E and G.

---

> ### Author Response · Authors · 2020-11-14
> **Answer to AnonReviewer2, question 1**
>
> > In the paper, the introduced disentanglement learning method can split information captured from images into a task-specific feature and another task-orthogonal representation. However, the proposed method cannot consider the domain-specific and domain-orthogonal information, which may degrade the performance in experiments and deserves to be discussed further. So how to learn the domain-invariant representation in the model? Please explain more details.
>
> DiCyR is not explicitly trained to produce domain-invariant representations as can be done in other mentioned works using a domain discriminator. DiCyR indirectly learns to produce domain-invariant representations thanks to the feature-level cyclic-consistency. Making sure that a generated cross-domain sample has the same task-related features as the original sample from the opposite domain allows the task-related latent space to be domain invariant. Our contribution can be interpreted as narrowing down this domain-independent information to the more precise task-specific information. We modified the last paragraph of 4.2 to explicitly specify this subtlety. Would this interpretation answer your question?

---

### Official Review · AnonReviewer3 · 2020-10-29
**The novelty is limited and the experiment of this paper is not strong enough.**

**Rating:** 4
**Confidence:** 3

**Review:**

This paper proposes a new framework for unsupervised domain adaptation by applying the disentangled representations learning (DiCyR). The core idea of DiCyR is to split the raw feature into the task-related one and its complimentary context where the task-related representations are projected into a shared space for alignment. From my point of view, disentangled representations learning is the most interesting part of this work.


-Pros:
1. This paper is well organized with clear logic to follow. The authors have introduced the problem statement and related works very well.
2. The motivation of disentangled representations learning for domain adaptation is very straightforward and clear.
3. The authors provide the demo code of this work to prove its reproductivity.
4. There are some visual results that could attract readers a lot.

-Cons:
1. Novelty:
Taking the cross-domain image translation is a smart way of data augmentation, which has been well studied. Since 2017, CyCADA has introduced the CycleGAN framework for domain adaptation and has received a significant impact since then. This work extends the image translation idea by applying the disentangled image generation based on cyclic consistency. Moreover, there are a lot of previous papers working on similar topics [1, 2, 3, 4]. In this way, I think the novelty of this paper is minor.

2. Experiment:
In the domain adaptation task, the authors only evaluate its model on the toy datasets (Digital Datasets), which cannot fully prove this model's effectiveness. I am curious why not using the latest office-home, domainnet, or the Cityscapes-to-GTA5 as the main experiment. On MNIST to SVHN, the DiCyR performs rather weakly. Moreover, there are more recent works, such as [5] should be compared.

3. Time and Computational Cost
CycleGAN-based image translation usually takes a lot of training time. The author mainly used the small size images (32 \times 32) for evaluation, and it requires a P100 GPU for training. What the time and computational cost will be if using large images such as 224 \times 224?

If the authors could address all the concerns above, I will consider upgrading the score.


[1] Cai R, Li Z, Wei P, et al. Learning disentangled semantic representation for domain adaptation. IJCAI, 2019.

[2] Cao J, Katzir O, Jiang P, et al. Dida: Disentangled synthesis for domain adaptation[J]. arXiv preprint arXiv:1805.08019, 2018

[3] Li H, Wan R, Wang S, et al. Unsupervised Domain Adaptation in the Wild via Disentangling Representation Learning. IJCV, 2020

[4] Yang, Junlin, et al. "Unsupervised domain adaptation via disentangled representations: Application to cross-modality liver segmentation." MICCAI, 2019.

[5] Hosseini-Asl, Ehsan, et al. "Augmented cyclic adversarial learning for low resource domain adaptation." ICLR, 2019.

---

> ### Author Response · Authors · 2020-11-14
> **Answer to AnonReviewer3 question 3**
>
> >Time and Computational Cost CycleGAN-based image translation usually takes a lot of training time. The author mainly used the small size images (32x32) for evaluation, and it requires a P100 GPU for training. What the time and computational cost will be if using large images such as 224x224
>
>
> We will include the complete computational cost assessment with the experiment mentioned above.

---

> > ### Author Response · Authors · 2020-11-23
> > **Requested additional experiments**
> >
> > As suggested, we ran an additional experiment on a large scale dataset: GTA5 -> Cityscapes. We assessed both the model performance and computational times. The results were included in the paper and detailed results were added to the Appendix, sections E and G.

---

> ### Author Response · Authors · 2020-11-14
> **Answer to AnonReviewer3 question 2**
>
> >Experiment: In the domain adaptation task, the authors only evaluate its model on the toy datasets (Digital Datasets), which cannot fully prove this model's effectiveness. I am curious why not using the latest office-home, domainnet, or the Cityscapes-to-GTA5 as the main experiment. On MNIST to SVHN, the DiCyR performs rather weakly. Moreover, there are more recent works, such as [5] should be compared.
>
>
> Thank you for this very relevant remark. We provided a new experiment on a new setting Syn-Signs→GTSRB. It is still a small images dataset but we believe that it provides more evidence of the model's effectiveness. We did not use the classically used Office dataset because it contains very few images by categories (less than 50 in average). Most of the works addressing this type of dataset use a pre-trained backbone such as ResNet50.
> We still believe that it is important to evaluate our method on larger datasets and are doing our best to provide a new experiment on the VisDA 2017 or the Cityscapes-to-GTA5 datasets.

---

> ### Author Response · Authors · 2020-11-14
> **Answer to AnonReviewer3 question 1**
>
>
> >Novelty: Taking the cross-domain image translation is a smart way of data augmentation, which has been well studied. Since 2017, CyCADA has introduced the CycleGAN framework for domain adaptation and has received a significant impact since then. This work extends the image translation idea by applying the disentangled image generation based on cyclic consistency. Moreover, there are a lot of previous papers working on similar topics [1, 2, 3, 4]. In this way, I think the novelty of this paper is minor.
>
> The cross-domain image translation introduced by CycleGAN has indeed generated a whole lot of research interest on its application to the domain adaptation problem.
> Thank you for pointing out references [3] and [4] that we missed previously.
> They are now all included in our manuscript.
> Similarly to the papers you mention, we assume that learning a disentangled representation is an efficient way to identify common representations between domains. However, we believe that our method differs in its intentions and the resulting properties of the learned representations:
>
> >[1] Cai R et al., Learning disentangled semantic representation for domain adaptation. IJCAI, 2019.
>
> Similarly to our work, the authors aim to learn a task-related and a task orthogonal representation. However, their setting uses target data only to train an adversarial domain discriminator (along with source data), while the adversarial classifier is trained on the source data exclusively. If such a setting does allow an efficient disentanglement for elements of the source domain, generalizing this property to elements from the target domain is not straightforwardly guaranteed.
>
> >[2] Cao J et al., Dida: Disentangled synthesis for domain adaptation. arXiv preprint arXiv:1805.08019, 2018.
>
> With a similar method to the one used in [1], the authors learn a disentangled representation that is used to generate cross-domain image translation. These augmented samples are then used to train the classifier in the target domain. Yet, the lack of cycle consistency in both the semantic and the feature level does not guarantee an efficient mapping between domains in the domain invariant latent space and could lead to generated samples associated with a wrong label.
>
> >[3] Li H et al., Unsupervised Domain Adaptation in the Wild via Disentangling Representation Learning. IJCV, 2020.
>
> The authors use a shared encoder and a shared generator to produce image-to-image translations. It also benefits from cyclic consistency to force the category-related information to be domain invariant.
> We argue that constraining a feature level cyclic-consistency on the task-related information is not sufficient to ensure that no information redundancy is present in one of the representations.
> The GRL layers we use to connect our feature predictors, associated with feature-level cyclic-consistency on both task and style features, avoid information sharing between the representations.
>
> >[4] Yang J et al., Unsupervised domain adaptation via disentangled representations: Application to cross-modality liver segmentation. MICCAI, 2019.
>
> The authors learn disentangled representation in both domains simultaneously using feature cyclic-consistency exclusively. A major problem remains the incentive to assign a constant encoding to one of the representations (the style information being represented through the generator) thus allowing eventually to maintain a cycle consistency at the feature level while failing at achieving separation between task-related and style information. Our work prevents this behavior from happening using a triplet loss with both positive and negative samples.
>
> While the intentions of our method are similar to those of the work mentioned above, we believe that our representations offer superior guarantees on the quality of the learned representations in both domains. We believe that it is this principled approach on the quality of the representations that differentiate our method from previous work and allows it to obtain better results than the other methods using cycle-consistency and disentanglement. Besides, our method is the only one to allow disentanglement in a single domain setting as illustrated by the experiments of section 5.1.

---

### Author Response · Authors · 2020-11-14
**Summary of changes**

We want to thank the reviewers for their attentive and helpful comments and their constructive suggestions that will allow us to improve the quality of our manuscript.
Below we respond to the comments of each reviewer in detail (for readability we quoted the question before answering). We are also providing a revised manuscript that reflects their suggestions and comments. We feel that this has resulted in a stronger manuscript.

Summary of actions taken:
- clarified some notions in the paper (changes in red),
- included results for the additional Syn-Signs→GTSRB experiment,
- improved results with better hyperparameters and related discussion,
- started the requested additional experiments.

---

### Decision · Program_Chairs · 2021-01-07
**Final Decision**

**Decision:**

Reject

**Comment:**

This paper investigates the problem of unsupervised domain adaptation and proposes a framework based on a specific type of disentangled representations learning. The paper is well written and the proposed method seems plausible. However, according to Reviewers #3 and #4, the proposed framework does not seems to be sufficiently different from existing ones, and the empirical results do not seem convincing enough.

Please also double check in C3, whether T and S should be marginally independent or conditionally independent conditioning on X.